# Greedy optimization of the geometry of Majorana Josephson junctions

André Melo⋆, Tanko Tanev and Anton R. Akhmerov

Kavli Institute of Nanoscience, Delft University of Technology, P.O. Box 4056,
2600 GA Delft, The Netherlands

⋆ am@andremelo.org

## Abstract

Josephson junctions in a two-dimensional electron gas with spin-orbit coupling are a promising candidate to realize topological superconductivity. While it is known that the geometry of the junction strongly influences the size of the topological gap, the question of how to construct optimal geometries remains unexplored. We introduce a greedy numerical algorithm to optimize the shape of Majorana junctions. The core of the algorithm relies on perturbation theory and is embarrassingly parallel, which allows it to explore the design space efficiently. By introducing stochastic variations in the junction Hamiltonian, we avoid overfitting geometries to specific system parameters. Furthermore, we constrain the optimizer to produce smooth geometries by applying image filtering and fabrication resolution constraints. We run the algorithm in various setups and find that it reliably produces geometries with increased topological gaps over large parameter ranges. The results are robust to variations in the optimization starting point and the presence of disorder, which suggests the optimizer is capable of finding global maxima.



# 1 Introduction

Majorana bound states (MBS) are topologically protected edge states with non-abelian statistics that can serve as a building block for fault tolerant quantum computers [1–4]. While much of the experimental search for topological superconductivity has focused on proximitized semiconducting nanowires [5–8], this system requires applying a sufficiently strong magnetic field to drive a topological phase transition. Because magnetic fields suppress superconductivity, the field compatibility of Majorana devices is an open problem. An alternative proposal—Josephson junctions in a two-dimensional electron gas (2DEG) with spin-orbit coupling [9–13]—uses the phase difference across the junction to lower the critical magnetic field.

The topological protection of MBS requires a spectral gap. Therefore, designing devices with sufficiently large gaps is a necessary component of engineering Majorana states. In Majorana Josephson junctions, the gap is limited by long trajectories in the normal region that do not come into contact with the superconducting terminals [14,15]. Eliminating these long-flight trajectories by making the junction zigzag-shaped leads to an order of magnitude increase in the topological gap [15]. The topological gap is also enhanced in other periodically modulated geometries [16,17].

Optimizing a band gap is similarly relevant to photonic and acoustic crystals to design devices such as filters, beam splitters, and waveguides [18]. A large body of research shows that numerical optimization methods such as genetic algorithms [19,20], semidefinite programming [21,22], and gradient-based strategies [23,24] find geometries with large band gaps despite performing a search in an exponentially large design space. More recent work demonstrated that deep learning accelerates optimization by predicting effective tight-binding models corresponding to microscopic geometries [25]. In the context of one-dimensional Majorana nanowires, Boutine et al. [26] and Turcotte et al [27] used an algorithm based on GRAPE [28] to minimize the Majorana localization length through spatially varying electrostatic potentials and magnetic field textures.

While geometry was demonstrated to have a sizeable effect on the topological gap of Majorana junctions, the question of how to find optimal geometries remains open. Inspired by the previous works in numerical geometry optimization, we develop the following greedy algorithm to find optimal Majorana junction geometries. At each optimization step, we compute a set of possible deformations to the shape of the superconducting regions. Using perturbation theory we estimate how the gap changes with these deformations and select the one that yields the largest improvement. We avoid overfitting geometries to specific parameters by randomly varying the operating point throughout the optimization, similarly to stochastic gradient descent. To ensure that the resulting geometries are within reach of fabrication techniques, we incorporate smoothness and minimum feature size constraints. We benchmark our algorithm on a variety of physical scenarios and find that it reliably produces geometries with increased topological gaps over large system parameter ranges. Finally, we check the robustness of the algorithm and discuss its potential generalizations.

# 2 Model and algorithm description

We consider a Josephson junction formed by proximitizing a Rashba 2DEG with two *s*-wave superconductors (Fig. 1). We model the central normal region with a two-dimensional Bogoliubov-de Gennes Hamiltonian

$$H_{\mathrm{N}} = \left( \frac{\hbar^2}{2m}(k_x^2 + k_y^2) - \mu_{\mathrm{normal}} + \alpha(k_y \sigma_x - k_x \sigma_y) \right) \tau_z + E_Z \sigma_x \,,$$

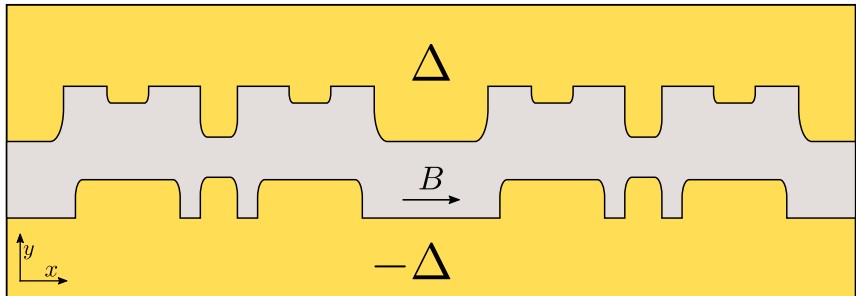

Figure 1: Schematic representation of two unit cells of a topological Josephson junction formed by covering a Rashba 2DEG with *s*-wave superconductors. An applied magnetic field penetrates the normal region (grey) and breaks time-reversal symmetry. The proximitized regions (yellow) experience an additional proximity-induced pairing term and a superconducting phase difference of $\pi$.

where $\alpha$ and $E_Z$ parameterize the strength of the Rashba spin-orbit and Zeeman fields respectively, $\mu$ is the chemical potential, and $\sigma_i$ and $\tau_i$ are the Pauli matrices acting in spin and electron-hole space. The proximitized regions experience an additional superconducting pairing interaction and expel the applied magnetic field, yielding the Hamiltonian

$$H_{\text{SC}} = \left( \frac{\hbar^2}{2m}(k_x^2 + k_y^2) - \mu_{\text{sc}} + \alpha(k_y \sigma_x - k_x \sigma_y) \right) \tau_z + \Delta(x, y) \tau_x.$$

Following previous works [9, 10] we fix the superconducting phase difference to its optimal value $\phi = \pi$, such that $\Delta(x, y) = \Delta_0$—the induced superconducting gap—in the top superconductor and $-\Delta_0$ in the bottom superconductor. This choice maximizes the area of the topological region in parameter space, practically guaranteeing that the system stays in the topological regime during the optimization. Because we are interested in determining bulk properties, we consider a translationally invariant system with a supercell Hamiltonian $H = H_N + H_S$, which we discretize using the Kwant software package [29]. Unless stated otherwise, we consider a lattice constant of $a = 20\,\text{nm}$, and Hamiltonian parameters $m = 0.02m_e$ (with $m_e$ the free electron mass), $\alpha = 20\,\text{meV nm}$, $\Delta_0 = 1\,\text{meV}$ and $\mu_{\text{normal}} = \mu_{\text{sc}} = \mu$.

Our core iterative algorithm starts by computing the band structure of the initial geometry on a sufficiently fine grid of the supercell momenta $\kappa_x$ using sparse diagonalization. Since we are interested in low-energy behavior, we compute only a small set of the $2n_b = 8$ bands closest to the Fermi level. We then compute a set of candidate perturbations to the geometry of the junction. Using conventional image processing we determine the normal-superconductor boundary and then consider two types of modifications: removing superconductivity from boundary sites, and introducing superconductivity in normal sites immediately next to the boundary (Fig. 2(b)). Limiting the geometry perturbations to the boundaries of the superconductors with the normal regions promotes the continuity of the shape evolution. In order to implement fabrication constraints, we reject perturbations where the minimum distance between superconductors is lower than a specified tolerance $w = 100\,\text{nm}$. Once we have collected a set of perturbations, we use first order degenerate perturbation theory to estimate how they change the dispersion relation (Fig. 2(c)). Finally, we modify the superconductors' shape with the perturbation that yields the largest improvement in the gap and proceed to another iteration (Fig. 2(d)). Because calculations of Bloch eigenstates at different $\kappa_x$ as well as computing the effect of different perturbations are independent, the algorithm is embarrassingly parallel. We leverage this by using the dask software package [30] to parallelize our implementation.

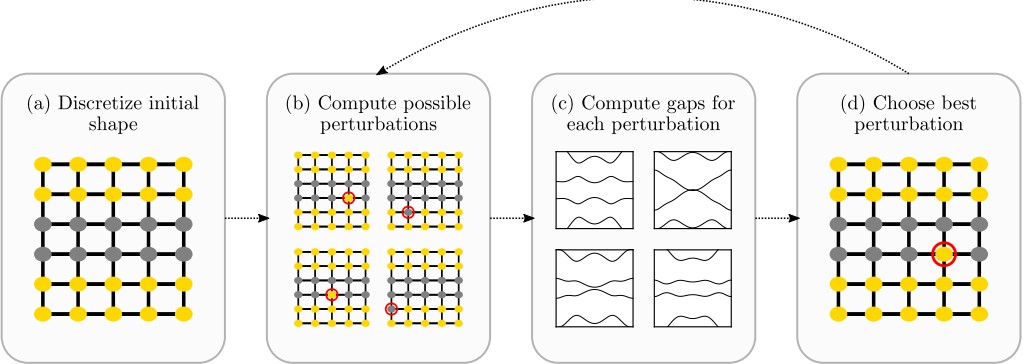

Figure 2: Summary of the core optimization algorithm. (a) We start from a discretized model of the junction. (b) At the start of an iteration we determine the boundaries of each superconductor. We then consider perturbations that introduce or remove superconductivity in sites at the boundary (circled in red). (c) Using perturbation theory, we compute how the gap changes with each geometry modification. (d) Finally, we update the shape using the modification that yields the largest improvement in the gap.

Although the core algorithm is already capable of finding improved geometries, it has the following limitations:

- It is inefficient to fully recompute the dispersion relation after each iteration.

- Due to being greedy, the algorithm gets stuck in any local maximum.

- The resulting shapes tend to be irregular and contain features that vary on the scale of the lattice constant.

We solve these problems by introducing epochs consisting of a handful of iterations each. Instead of exactly recomputing the dispersion relation at each iteration, we do it only at the beginning of an epoch. Furthermore, at every epoch we select random values of $E_{Z1} < E_Z < E_{Z2}$ and $\mu_1 < \mu < \mu_2$. This procedure is analogous to performing stochastic gradient descent to optimize the average gap over a region in parameter space, and it ensures that the optimized geometries are tolerant to variations in the junction Hamiltonian. Finally, we apply a median filter to the superconductor shapes every few epochs, a standard technique in image processing that reduces noise in images by replacing a pixel with the median value of its neighbors. As we will show in the next section, periodically applying this filter constrains the optimizer to explore shapes that vary smoothly in space.

## 3 Results

Having introduced the algorithm, we turn to investigate its performance, robustness, and the relevance of its components. Unless noted otherwise we perform epochs with 5 iterations per epoch and apply a median filter with a window size of $3a = 60$nm. Reflection symmetry $x \to L_x - x$ protects the gap from closing at finite momentum [31]. Therefore we consider only mirror-symmetric geometries $\Delta(x, y) = \Delta(L_x - x, y)$ and change the shape in pairs of reflection-symmetric sites simultaneously.[1]

---

[1] We have verified that the first order perturbative estimate of the gap after adding or removing two superconducting sites is always within 3% of the exact value in the systems we study.

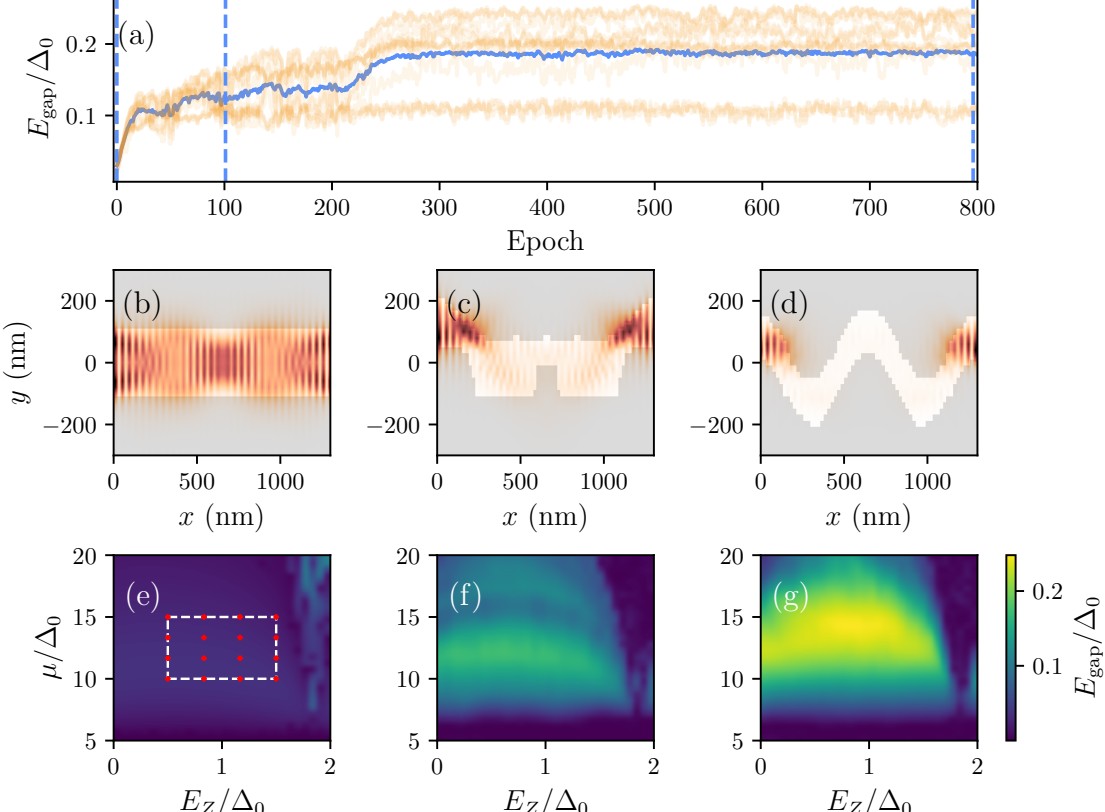

Figure 3: Optimization run starting from a straight geometry with homogeneous chemical potential and with parameter shifting and median filtering. (a) The blue curve corresponds to the average gap evaluated at 16 different points in parameter space, whose individual values we plot in orange. In panel (e) we mark these points with red dots. (b-d) Geometries found at the epochs marked with vertical dashed lines in panel (a). (e-f) Topological phase diagrams of the geometries in (b-d).

Figure 3 demonstrates our algorithm in action. We initialize the system with a straight geometry where the normal and superconducting regions have initial widths $W = 200\,\mathrm{nm}$ and $L_{\mathrm{SC}} = 500\,\mathrm{nm}$ respectively. Further, we fix the unit cell length at $L_x = 1300\,\mathrm{nm}$ (see App. A for results of optimization runs at other values). We consider a parameter space of $\mu_1 = 10\,\mathrm{meV}$, $\mu_2 = 15\,\mathrm{meV}$, $E_{Z1} = 0.5\,\mathrm{meV}$, $E_{Z2} = 1.5\,\mathrm{meV}$, and apply the median filter every 5 epochs. In Fig. 3(a) we show how the gap of 16 uniformly spaced points in the parameter space evolves with epoch number. Although the gaps at fixed parameters fluctuate, the average increases smoothly as the optimization proceeds, eventually converging to a value approximately 10 times larger than the initial one. In Fig. 3(b-d) we show geometries and corresponding Majorana wavefunctions at the beginning of the optimization, its middle, and at convergence. Similar to the zigzag geometries explored in [15], the final shape has a strong spatial modulation that eliminates long quasiparticle trajectories. Despite the algorithm imposing no such constraints, the converged geometry has a periodicity of half of the supercell. The increase in average gap leads to more localized Majorana wavefunctions, as is expected from the well-known relation for the localization length $\xi_M = \frac{\hbar v_F}{E_{\mathrm{gap}}}$. Indeed, in the last iteration, the overlap of the Majoranas is already negligible within a single supercell. In Fig. 3(e-g) we plot topological phase diagrams as a function of $\mu$ and $E_Z$ and find that the algorithm significantly increases the average gap both within the search window and in its vicinity.

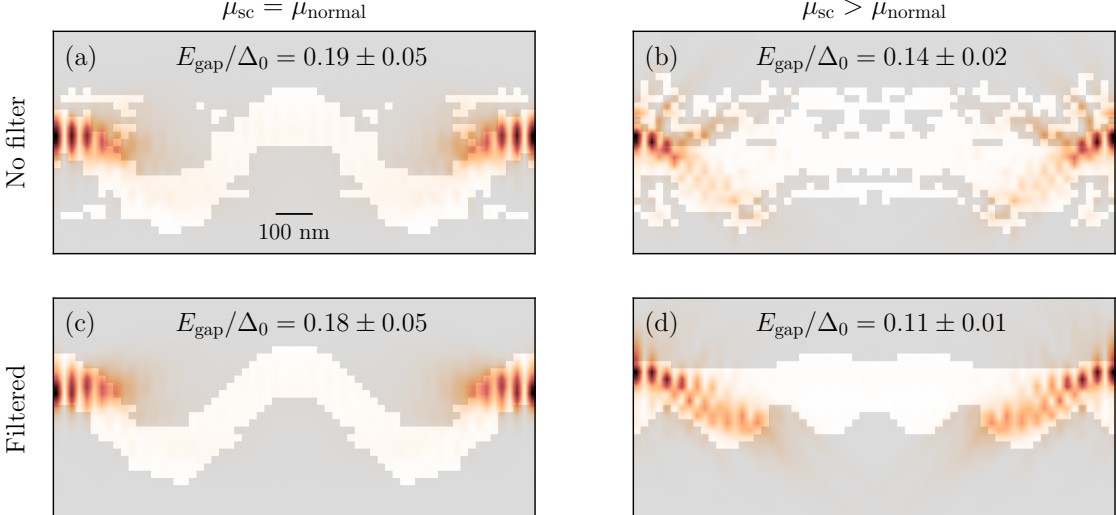

Figure 4: The effect of introducing median-filtering in the optimization algorithm. The top panels (a) and (b) show optimized geometries when starting from a straight junction with homogeneous and mismatched chemical potentials, respectively. The bottom panels (c) and (d) show the geometries obtained when a median filter is added. Each panel also shows the average gap over the range of $\mu$ and $E_Z$ used in the optimization (see main text for exact parameters).

We study the importance of individual aspects of the algorithm by examining their impact on the algorithm performance. To confirm the importance of the median filter, we check that excluding it generates a discontinuous shape shown in Fig. 4(a). Turning to the random sampling of parameters, we repeat the previous run with fixed parameters $\mu = 10\,\text{meV}$ and $E_Z = 1\,\text{meV}$, along with a parameter-shifting run with $\mu_1 = 9.5\,\text{meV}, \mu_2 = 10.5\,\text{meV}$, $E_{Z1} = 0.95\,\text{meV}, E_{Z2} = 1.05\,\text{meV}$.[2] We plot the observed gaps at each epoch in Fig. 5. Both runs result in geometries with significantly larger gaps than the straight junction. However, the optimizer with fixed parameters gets stuck in a local maximum with a gap of approximately $0.2\Delta_0$. In contrast, the parameter-shifting optimizer avoids this local maximum and continues exploring the geometry space until it converges to a shape with a gap of the order of $0.25\Delta_0$.

All of the previous simulations converged to the same geometry, which raises the natural question of whether the optimizer generalizes well to other physical scenarios. To address this, we consider a modified version of the 2DEG junction with larger chemical potential in the proximitized regions $\mu_{\text{sc}} = 15\,\text{meV}$, and a parameter range of $\mu_{\text{normal},1} = 9.5\,\text{meV}$, $\mu_{\text{normal},2} = 10.5\,\text{meV}, E_{Z1} = 1.35\,\text{meV}, E_{Z2} = 1.65\,\text{meV}$. The higher chemical potential in the proximitized regions simulates the doping due to the work function difference [32,33]. Figure 4(b) shows the geometry obtained with a run of 800 epochs without filtering. Interestingly, the optimizer converges to highly disordered superconductor shapes. This is in contrast with the previous simulations, where the unfiltered geometries remain approximately smooth. Applying the median filter at every epoch restores the smoothness of the final geometry (Fig. 4(d)), but results in a significant reduction in the average topological gap. We attribute the overall lower gaps to the presence of normal scattering at the normal-superconductor interface caused by the Fermi velocity mismatch [34,35]. This tends to reduce the induced superconducting gap and complicate the optimization problem. We have briefly explored optimization in another

---

[2]We choose a parameter window smaller than in Fig. 3 to reduce the fluctuations of the gap across epochs. In our simulations we found that the performance of the algorithms depends weakly on the window size, as long as it is neither too big nor too small.

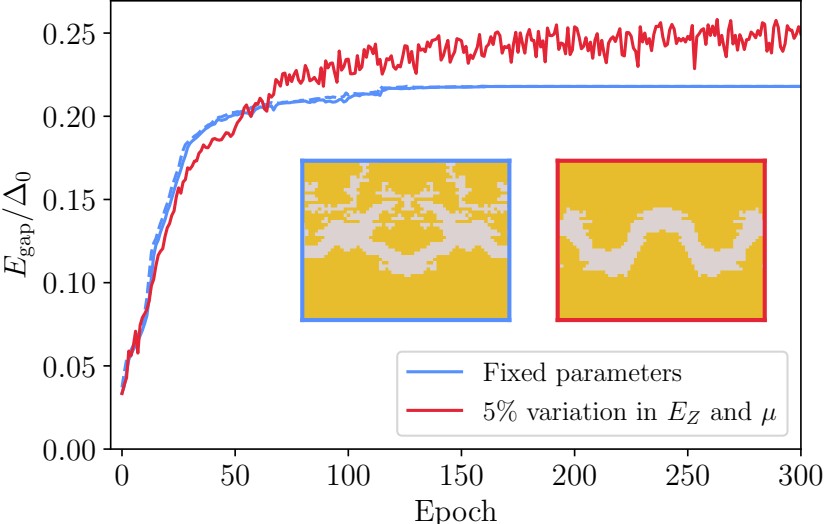

Figure 5: A comparison of the behavior of the optimizer with fixed Hamiltonian parameters and with random fluctuations in the chemical potential and Zeeman field. The blue curve shows the exact topological gap computed at each epoch for the optimizer with fixed parameters. At approximately 130 epochs, the optimizer gets stuck in a local maximum. In contrast, the optimization with parameter shifting (red curve) avoids this local maximum and converges to higher gap values. Although the geometry has converged, the observed gaps oscillate about their average value due to the parameter shifting. The two inset plots show the converged geometries obtained at each run.

setup consisting of a straight junction with electrostatic gates next to the superconductors. Such a system would be less influenced by diamagnetic screening supercurrents that suppress the amplitude of Andreev reflection amplitude and hence the induced gap [36]. However, we did not obtain systematic results yet and thus we omit the discussion of these systems from the manuscript.

To study the robustness of our algorithm, we introduce several modifications in the optimization procedure and study how the results change in comparison with the reference geometry from Fig 3. In Fig. 6(a-b), we show results obtained with a different random seed and in Fig. 6(c-d) when the optimization starts from a zigzag geometry [15] (width and amplitude modulation of $W = z_y = 300$ nm). In both scenarios, the optimization converges to the reference up to 1–2 lattice sites. This suggests that the final result is independent of the details of the simulation parameters and that it is likely that the algorithm is converging to a global maximum in geometry space. Next, we allow the algorithm to add a single site per iteration, thereby removing the mirror symmetry constraint. To maintain consistency in the number of sites added per epoch we perform 10 iterations. Remarkably, although the optimizer starts by exploring highly mirror-asymmetric configurations, it eventually converges to the mirror-symmetric reference shape (Fig. 6(d-e)). Finally, we explore the effects of disorder in the junction Hamiltonian. We introduce an onsite potential $V_{\mathrm{disorder}}(x, y) = V_0(x, y)\tau_z$, where $V_0(x, y)$ is uniformly distributed in $[-W_0, W_0]$. We set the disorder strength at $W_0 = 1.7$ meV, which corresponds to a mean free path of a unit cell length. To avoid overfitting to a specific disorder realization we sample a new set of $V_0(x, y)$ every epoch. Disorder effectively renormalizes Hamiltonian parameters and hence plays a similar role to parameter shifting. Therefore we opt for a smaller parameter window and set

$\mu_1 = 9.5\,\mathrm{meV}, \mu_2 = 10.5\,\mathrm{meV}, E_{Z1} = 0.95\,\mathrm{meV}, E_{Z2} = 1.05\,\mathrm{meV}$. Once again the resulting shape differs from the reference by a few lattice sites (Fig. 6(g-h)).

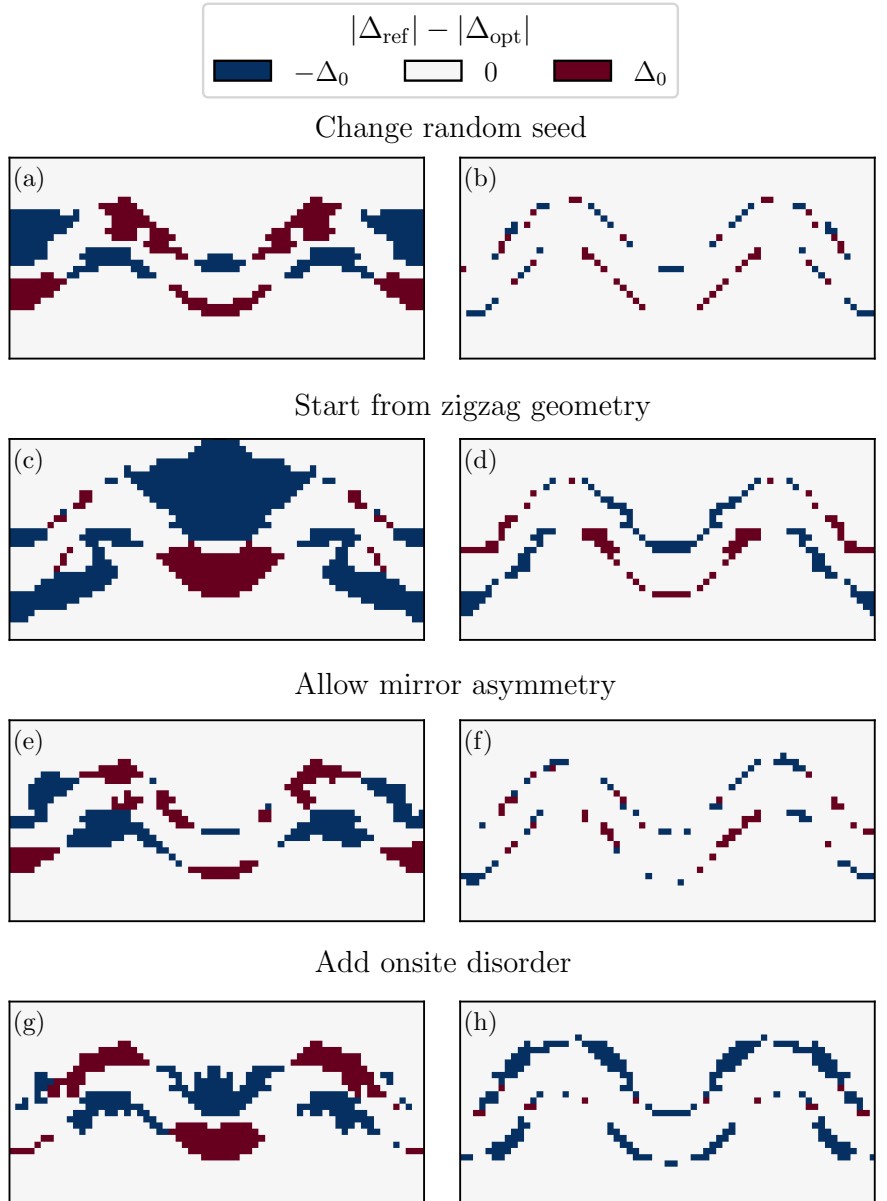

Figure 6: Comparison of optimized geometries with a reference geometry obtained by optimizing a straight junction with homogeneous chemical potential. In the panels we plot the difference of the magnitude of the superconducting gap of the reference geometries and the optimized geometries. The left panels show the geometry obtained after 50 epochs, and the right panels after convergence. Panels (a-b) correspond to geometries obtained with a different random seed, (c-d) starting the optimization from a zigzag shape, (e-f) allowing the optimizer to choose mirror-asymmetric perturbations, and (g-h) in the presence of onsite disorder.

# 4 Summary and outlook

We have presented a greedy algorithm for finding Majorana junction geometries with large topological gaps. Our algorithm relies on parallelization and perturbation theory to sample geometry space efficiently and is compatible with minimum feature size and smoothness constraints. We validated our approach in different scenarios and showed that it is robust to variations in the starting point and the presence of disorder.

There are several possible improvements to the algorithm. A straightforward optimization would be to explore the hyperparameter space more systematically and find, for example, the optimal number of iterations per epoch and parameter window size. Additionally, the optimized geometries presented in the main text have periods shorter than our initial choice of $L_x$. This suggests that allowing the optimizer to dynamically adjust the period of the unit cell may increase performance. Another direction of further research would be to go beyond our greedy strategy and implement more sophisticated algorithms to explore the tree of perturbations such as Monte Carlo tree search [37].

While our results look promising, they are not yet experimentally relevant. The state-of-the-art experiments in 2DEG heterostructures do not offer sufficient insight into the material properties to enable predictive simulations. On the theoretical side, we have neglected several phenomena, namely electrostatics and magnetic field distribution, which will strongly influence the optimal geometry shape. Including these effects in the simulation requires significantly higher computational resources and is not justified without more knowledge about the platform.

We expect that our algorithm applies to other Majorana devices, such as Majorana Josephson junctions that only require phase gradients to break time-reversal symmetry [38, 39]. Geometry optimization can answer whether the previously reported small gaps in the high-density regime are an inherent problem of this platform. While our initial experiments indicate that the algorithm is directly suitable to gate shape optimization, more work is required to achieve a reliable conclusion.

Considering geometry optimization beyond Majorana Josephson junctions, we believe that the core ideas of our approach would apply to other inverse design problems in mesoscopic quantum physics [40]. The stochastic nature of our algorithm safeguards it from overfitting, but on the other hand makes it unsuitable to find sharp resonances or phenomena that are sensitive to microscopic device details. This, however, is a natural setting in many experiments where the control over the system is imprecise. The numerical efficiency of our approach largely relies on the locality of the perturbation used to estimate the gradient of the target function. Local control over microscopic Hamiltonians is far beyond current experimental reach. A practical adaptation of our algorithm to nonlocal perturbations would approximate those as local during most optimization steps and only recompute the precise observables at the beginning of an epoch.

# Acknowledgements

We are grateful to Mert Bozkurt, Evert van Nieuwenburg and Andrew Saydjari for useful discussions.

**Data availability**   The code used to generate the figures is available on Zenodo [41].

**Author contributions**   A.A. defined the goal of the project, and A.A. and A.M. designed the approach. T.T. implemented an initial unreported version of the optimization under the super-

vision of A.M. and A.A. A.M. implemented the final version of the optimizer and performed the numerical experiments in the manuscript. A.M. and A.A. interpreted the results. A.M. wrote the manuscript with input from A.A.

**Funding information**   This work was supported by the Netherlands Organization for Scientific Research (NWO/OCW), as part of the Frontiers of Nanoscience program and an NWO VIDI grant 016.Vidi.189.180.

## A   Investigating the role of the supercell size

In the optimization runs reported in the main text we kept the unit cell period fixed at $L_x = 1300$ nm and found that most runs converged to a geometry with a period of 650 nm. To investigate the role of this parameter in the optimization, we repeat the runs of Fig. 3 with different values of $L_x$. In Fig. 7(a-b) we show optimized geometries for unit cell periods of $L_x = 650$ nm and 1950 nm. Interestingly, the resulting shapes are similar to those of Fig. 3 and have the same period of 650 nm. In Fig. 7(c-d) we show results for two additional runs at $L_x = 975$ nm and 1625 nm. Although the resulting shapes are qualitatively similar to the previous results, the optimizer is forced to choose a different period because the unit cell lengths are incommensurate with 650 nm.

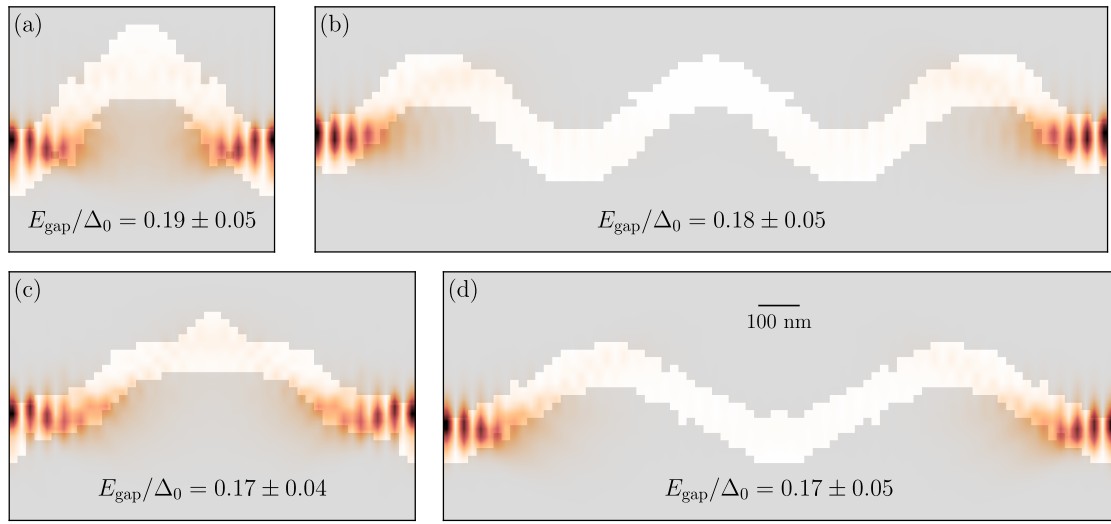

Figure 7: Optimized geometries starting from supercell sizes of (a) 650 nm, (b) 1950 nm, (c) 975 nm, and (d) 1625 nm. The remaining optimization and Hamiltonian parameters are identical to those of Fig. 3. Each panel also shows the average gap over the range of $\mu$ and $E_Z$ used in the optimization (see main text for exact parameters).

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
