# Peer review of "Greedy optimization of the geometry of Majorana Josephson junctions"

_SciPost Physics, doi:SciPost Phys. 14, 047 (2023)_

## Round 2 · Referee Report · Anonymous · 2022-7-29

Strengths

1) New approach to the problem (see the report) - greedy optimisation instead of ad hoc design of the MJJ device.
2) An effective strategy - perturbative treatment which allows for an embarrassingly parallel algorithm

Weaknesses

1) Little practical guidance for experimentalists (beyond the already known facts: the JJ should not be straight, minimal width etc.) unless concrete materials are considered - see the end of report
2) It is not clear if this way of optimising the geometry offers a significant advantage over any kind of "straightforward educated guess" (see the reference to PRL 125, 086802 in report)

Report

Melo et al. report a theoretical study of geometry optimisation of MJJ (Marjorana Josephson junction), see strenghts #1 It is known that straight MJJs suffer from superconducting gap reduction as reported previously by some of the Authors in ref. 15 (on the other hand, see weaknesses #2); results of the optimisation procedure presented in the manuscript suggest that the zig-zag structure (such as seen in Fig. 5 for example) mitigates this shortcoming the best. Is this outcome dependent on the model geometry? (If a rectangle with different aspect ratio were chosen, would the "wavelength" of the zig-zag line remain the same?) If the optimal MJJ design is generic, how does it depend on the model parameters? (alpha and Delta in particular) Also, the validity of perturbative treatment could be supported by some illustrative example (I don't have any reason to doubt it but seeing is believing; also constraints on the validity of this approach could be specified).

In general, research reported seems solid but of somewhat limited usefulness - combining the proposed modelling with fabrication of actual devices (and confirming the superior properties of optimised design) would be a game changer, of course. Also, addressing #2 of weaknesses could increase the impact of the present research.

Requested changes

1) consider the suggestions mentioned in report
2) rearrange the figures in a way which corresponds to the actual text flow

  • validity: good
  • significance: good
  • originality: high
  • clarity: high
  • formatting: good
  • grammar: excellent

Author:  André Melo  on 2022-11-01  [id 2970]

(in reply to Report 1 on 2022-07-29)

We thank the referee for taking the time to review the manuscript and for their useful comments.

1) Little practical guidance for experimentalists (beyond the already known facts: the JJ should not be straight, minimal width etc.) unless concrete materials are considered - see the end of report

We agree with this remark, and we believe this is already accurately addressed in our conclusions. We do, however, care to use reasonable material parameters in our simulations. Given that the current knowledge about the material physics of these platforms is limited, we believe it would be premature to perform more detailed simulations. As knowledge of the material properties of hybrid nanostructures continues to improve, we expect that our algorithm (possibly combined with more detailed simulations) will become practically useful.

2) It is not clear if this way of optimising the geometry offers a significant advantage over any kind of "straightforward educated guess" (see the reference to PRL 125, 086802 in report)

We disagree with the referee on this particular point. While it is true that the shapes we present resemble the previously reported zigzag geometry, the resulting gaps are approximately a factor of 2 larger than the values reported in the zigzag paper. Moreover, we have demonstrated that changing the physical system (in particular by adding a density mismatch) results in a different shape that is quite different from the zigzag geometry (Fig. 4d of the manuscript). Finally, the algorithm we present is quite general and we believe it would be useful in other optimization tasks where physical intuition is more scarce (e.g. optimizing the geometry of braiding setups).

The referee also writes:

(If a rectangle with different aspect ratio were chosen, would the "wavelength" of the zig-zag line remain the same?)

We refer to our reply to Report #2 where we have addressed this question.

If the optimal MJJ design is generic, how does it depend on the model parameters? (alpha and Delta in particular)

The optimal MJJ design depends on material parameters, as demonstrated by the simulation where we introduce a density mismatch. As the referee suggests, mapping out the optimal designs as a function of Hamiltonian parameters would be interesting. However, because our primary goal is to introduce the optimization algorithm and demonstrate its validity, and because such analysis would be premature (see the explanation above and in the conclusion of the manuscript), we choose to leave this investigation for further work.

Also, the validity of perturbative treatment could be supported by some illustrative example (I don't have any reason to doubt it but seeing is believing; also constraints on the validity of this approach could be specified).

We have added a clarifying footnote to this end:

We have verified that the first-order perturbative estimate of the gap after adding or removing two superconducting sites is always within 3% of the exact value in the systems we study.

We also attach a plot that shows the perturbative estimate vs the actual gap below.

Following the referee's comments, we also rearranged the results sections and the corresponding figures to improve textual flow.

Attachment:

---

## Round 2 · Referee Report · Anonymous · 2022-8-10

Strengths

1) Fairly general optimization scheme for the problem

2) Detailed examination of the robustness of the results

Weaknesses

1) Idealized model, unclear if detailed optimization will reflect in experiments

Report

The manuscript describes a stochastic numerical method for optimizing 2D geometry of planar Josephson junction, maximizing the MBS energy gap in the system.

The form of the optimal shape is not a priori obvious, aside from the need of blocking long trajectories, and in principle this question is interesting. The effect of the shape in the system here was considered in previous works by some of the authors, but not systematically. The main new point in the descent algorithm, similar to T=0 simulated annealing, appears to be the choice of "good" neighborhood structure for the problem here and exploring robustness of its performance. Similar ideas can be useful when considering related mesoscopic structure optimization problems.

The experimental relevance of the results are not clear, as the authors also remark. The precise shape is probably not a main factor limiting realization of the system. In the examples in this work, detailed optimization does not appear to provide orders of magnitude increase (eg. 50% gap increase is seen between Fig. 3c/3d).

On p.5 it is found the optimal shape converges to a periodicity of half of the supercell. Is this a general feature that is independent of the supercell size. Is the supercell size choice here arbitrary? What do the results here imply for the optimal shapes in finite structures, should they be similar?

In principle the direction studied here is interesting, even though the practical impact may be limited. The manuscript could be suitable for SciPost Physics, after the authors consider the above comments.

Requested changes

Explain how the supercell size was chosen (arbitrary?), and whether the
half-supercell periodicity is generic.

The text should remark earlier that chemical potential and other parameters aside from Delta are assumed to be the same in proximitized and unproximitized parts.

  • validity: high
  • significance: ok
  • originality: high
  • clarity: high
  • formatting: excellent
  • grammar: excellent

Author:  André Melo  on 2022-11-01  [id 2968]

(in reply to Report 2 on 2022-08-10)

We thank the referee for taking the time to review the manuscript and for their useful comments.

On p.5 it is found the optimal shape converges to a periodicity of half of the supercell. Is this a general feature that is independent of the supercell size. Is the supercell size choice here arbitrary? What do the results here imply for the optimal shapes in finite structures, should they be similar?

For the main demonstration of the algorithm, we chose a supercell size comparable to existing devices, such that our results are directly applicable to finite structures. Systematically exploring different supercell sizes $L_x$ would be a relevant investigation. We also allude to this in our conclusions, where we propose to modify the algorithm to dynamically adjust the supercell size. As a first step in this direction, we have added an appendix where we show results of optimization runs at different unit cell periods. We observe that the optimization algorithm converges to a different number of periods within the system size, so that the period of the optimal structure stays approximately the same.

Finally, as suggested by the referee we have explicitly pointed that the chemical potential is taken to be homogeneous unless stated otherwise:

Unless stated otherwise, we consider a unit cell of length $L_x = 1300 nm$, lattice constant of $a = 20 nm$, and Hamiltonian parameters $m = 0.02m_e$ (with $m_e$ the free electron mass), $\alpha = 20 meV / nm$, $\Delta_0 = 1 meV$ and $\mu_\mathrm{normal} = \mu_\mathrm{sc} = \mu$.

---

## Round 2 · Referee Report · Anonymous · 2022-8-11

Strengths

1) The question of how to optimize topological Josephson junctions is an interesting one, which is also shown by the appearance of recent preprint arXiv:2208.02182 where a similar question is addressed.

2) I find that the presentation of the method and the explanation of the results are clear. The authors address important issues, such as overfitting or the required smoothness for the obtained geometry.

Weaknesses

I have only a couple of minor remarks:

1) The authors mention that “ we apply a median filter to the superconductor shapes”. I am not sure how obvious is what is meant by a “median filter”, maybe it could be described in a few words.

2) I have the impression that the references to the figures are not adequate at certain places. For example:
i) on page #4 the authors write “...Figure 5 demonstrates our algorithm in action.” I think they want to refer to Fig.3 ?

ii) on page #5 the authors write “...To confirm the importance of the median filter, we check that excluding it generates discontinuous shapes shown in Fig. 5(a) and (c)”. I can see discontinuous shapes in Fig.5(a) and Fig.5(b).

iii) on page #6 the authors write “In Fig. 6(a-b), we show results obtained with a different random seed and in Fig. 6(a-b) when the optimization starts from a zigzag geometry [15]... “ I suppose the second reference should be Fig. 6(c-d) in this sentence, as suggested by the caption of Fig.6.

Report

Apart from the above minor remarks, I can recommend the publication of the manuscript as it is. I believe that the acceptance criteria of this journal are met.

  • validity: high
  • significance: high
  • originality: high
  • clarity: top
  • formatting: excellent
  • grammar: perfect

Author:  André Melo  on 2022-11-01  [id 2967]

(in reply to Report 3 on 2022-08-11)

We thank the referee for taking the time to review the manuscript and for their useful comments. To address the feedback given in the report we have made the following changes:

  1. To explain how the median filter works we added this clarification:

    Finally, we apply a median filter to the superconductor shapes every few epochs, a standard technique in image processing that reduces noise in images by replacing a pixel with the median value of its neighbors. As we will show in the next section, periodically applying this filter constrains the optimizer to explore shapes that vary smoothly in space.

  2. We have corrected the figure numbering throughout the main text.

---

## Round 3 · Author Response

See replies to referee reports.

---

## Editorial Decision

published